# Leptin induces TNFα-dependent inflammation in acquired generalized lipodystrophy and combined Crohn's disease

Jörn F. Ziegler[1,2,13], Chotima Böttcher [1,3,13], Marilena Letizia[1,2], Cansu Yerinde[1,2], Hao Wu[1,2], Inka Freise[1,2], Yasmina Rodriguez-Sillke [1,2], Ani K. Stoyanova[1,4], Martin E. Kreis[1,4], Patrick Asbach[1,5], Desiree Kunkel [1,6], Josef Priller[1,3,7], Ioannis Anagnostopoulos[1,8], Anja A. Kühl[1,9], Konstanze Miehle[10], Michael Stumvoll[10], Florian Tran[11], Broder Fredrich[11], Michael Forster [11], Andre Franke [11], Christian Bojarski[1,2], Rainer Glauben[1,2], Britt-Sabina Löscher [11], Britta Siegmund [1,2,13]* & Carl Weidinger [1,2,12,13]*

Leptin has been shown to modulate intestinal inflammation in mice. However, clinical evidence regarding its immune-stimulatory potential in human Crohn's disease remains sparse. We here describe a patient with the unique combination of acquired generalized lipodystrophy and Crohn's disease (AGLCD) featuring a lack of adipose tissue, leptin deficiency and intestinal inflammation. Using mass and flow cytometry, immunohistochemistry and functional metabolic analyses, the AGLCD patient was compared to healthy individuals and Crohn's disease patients regarding immune cell composition, function and metabolism and the effects of recombinant N-methionylleptin (rLeptin) were evaluated. We provide evidence that rLeptin exerts diverse pro-inflammatory effects on immune cell differentiation and function, including the metabolic reprogramming of immune cells and the induction of TNFα, ultimately aggravating Crohn's disease in the AGLCD patient, which can be reversed by anti-TNFα therapy. Our results indicate that leptin is required for human immune homeostasis and contributes to autoimmunity in a TNFα-dependent manner.

[1] Charité–Universitätsmedizin Berlin, Corporate Member of Freie Universität Berlin, Humboldt-Universität zu Berlin and Berlin Institute of Health, Berlin, Germany. [2] Department of Gastroenterology, Infectious Diseases and Rheumatology, Campus Benjamin Franklin, Berlin, Germany. [3] Laboratory of Molecular Psychiatry and Department of Neuropsychiatry, Berlin, Germany. [4] Department of Visceral Surgery, Campus Benjamin Franklin, Berlin, Germany. [5] Department of Radiology, Campus Benjamin Franklin, Berlin, Germany. [6] BIH Cytometry Core, Berlin Institute of Health, 10178 Berlin, Germany. [7] BIH Berlin, DZNE Berlin and University of Edinburgh and UK DRI, Edinburgh, UK. [8] Department of Pathology, Campus Charité Mitte, Berlin, Germany. [9] iPATH. Berlin–Immunopathology for Experimental Models, Core Facility of the Charité, Berlin, Germany. [10] Medical Department III-Endocrinology, Nephrology, Rheumatology, University of Leipzig Medical Center, Leipzig, Germany. [11] Institute of Clinical Molecular Biology, Christian-Albrechts-University of Kiel, Kiel, Germany. [12] Clinician Scientist Program, Berlin Institute of Health, Berlin, Germany. [13] These authors contributed equally: Jörn F. Ziegler, Chotima Böttcher, Britta Siegmund, Carl Weidinger. *email: britta.siegmund@charite.de; carl.weidinger@charite.de

The adipokine leptin regulates the differentiation, function and metabolism of a variety of immune cell subpopulations, as well as of intestinal epithelial cells[1–3]. Likewise, leptin has been implicated in the pathogenesis of intestinal inflammation in Crohn's disease (CD), in which hyperplastic mesenteric fat ("creeping fat") wraps inflamed small intestinal segments[4], and acts as a source of leptin and additional adipokines, that can modulate both systemic immune cell composition, as well as intestinal epithelial cell function in animal models of colitis[5]. Previous studies have demonstrated that leptin deficiency as well as the pharmacologic blockade of the leptin receptor attenuate disease severity in mouse models of colitis[6,7], highlighting a potential role for leptin in inflammatory bowel diseases. Accordingly, leptin was shown to induce the proliferation and polarization of CD4$^+$ T helper (Th) cells in animal models of autoimmune diseases[8–11], whereas it suppresses the development and maintenance of regulatory CD4$^+$ T cells[12]. Furthermore, rectal application of leptin promotes intestinal inflammation in mice by activation of the NF-κB pathway in epithelial cells, suggesting that direct effects of leptin on the epithelium might contribute to the induction of inflammation in CD as well[13].

It recently became evident that the function and metabolism of immune cells are closely interconnected. Increased glucose metabolism is required for effector functions such as cytokine production in pro-inflammatory M$_1$-like macrophages and T cells, whereas regulatory T cells and immune-suppressive M$_2$-like macrophages highly depend on lipid oxidation-based metabolism[14]. Interestingly, leptin has recently been shown to orchestrate immune cell metabolism and function as it facilitates anaerobic glycolysis in murine Th17 cells resulting in increased production of IL-17 and neuronal inflammation in the model of experimental autoimmune encephalomyelitis, emphasizing a critical role for leptin at the crosslink of metabolism and function of immune cells[15].

However, human data on the immune modulatory effects of leptin are limited to extremely rare diseases including (i) congenital leptin deficiency where T cell hypo-responsiveness, as well as metabolic dysfunction have been shown to be reversible by leptin substitution[16], and (ii) acquired generalized lipodystrophy[17], a condition with approximately 100 known cases worldwide[18]. Lipodystrophy patients suffer from a varying degree of adipocyte loss, resulting in hypo-leptinemia, severe insulin resistance, fatty liver and muscular degeneration. Acquired lipodystrophy can manifest at different ages and loss of adipose tissue can be complete and has been associated with autoimmunity[19]. To our knowledge, no other case of acquired generalized lipodystrophy and combined Crohn's disease (AGLCD) has previously been reported. Since the formation of the aforementioned creeping fat is restricted to small intestinal CD and cannot be observed in mouse models[20], little is known about the effects of creeping fat-derived leptin on immune cell differentiation and disease activity in CD.

Here, we describe a 21-year-old Caucasian male with AGLCD who received daily injections with 2.5 mg recombinant N-methionylleptin (rLeptin). Deep immune profiling by mass and flow cytometry and ex vivo functional assays in addition to clinical assessment before and after rLeptin substitution revealed distinct pro-inflammatory effects of rLeptin treatment that result in an aggravation of intestinal inflammation in a TNFα-dependent manner.

## Results

### Clinical phenotype of the AGLCD patient.
To date, it remains elusive how mesenteric fat and fat-derived leptin shape systemic inflammation in CD and very few clinical scenarios exist, in which the administration of recombinant adipokines is justified and in which their immune modulatory function can be studied in human pathophysiology. The AGLCD patient featured a complete lack of visceral and subcutaneous adipose tissue (Fig. 1a), absent leptin production (Fig. 1b), and severe intestinal inflammation (see Supplementary Note 1 and Supplementary Table 1 for a detailed case report and a summary of autoimmune serum markers of the AGLCD patient). Of note, none of the patient's relatives displayed similar symptoms (Fig. 1c) and no causative mutation could be detected in genes commonly associated with generalized lipodystrophy by exome sequencing (data not shown). The AGLCD patient had initially developed acquired generalized lipodystrophy at the age of 4 years and had been diagnosed with CD (Montreal Classification A1 L3 B2 + B3p) at the age of 11 years, thus suffering from an aggressive variant of CD. As a consequence of generalized lipodystrophy, the AGLCD patient had subsequently developed fatty liver degeneration requiring liver transplantation at the age of 15 years. Figure 1d summarizes the clinical history of the AGLCD patient and compiles lipodystrophy-related and Crohn's-related interventions and complications.

### Altered immune cell composition in the AGLCD patient.
In order to characterize how the absence of fat tissue affects systemic immune cell composition in the AGLCD patient and to obtain a baseline of his immunologic makeup, we first compared peripheral blood mononuclear cells (PBMCs) of the AGLCD patient with lymphocytes of healthy donors (HD) and patients with CD using mass cytometry for a high dimensional immune cell analysis which allowed us to better discriminate between lipodystrophy-associated and CD-specific immune cell alterations. Similar to our previously published protocol[21], antibodies against lineage markers for T cells (CD3, CD4, CD8), monocytic cells (CD11b, CD11c, CD14, EMR1), B cells (CD19) and NK cells (CD16, CD56), as well as antibodies against functional makers (CD36, CD163, TREM2, arginase1, CD206), differentiation markers (CD33, CD40, CD45, CD64, CD95, CD115, CD116, CD135), homing markers (CD54, CD68, CD103, CCR2, CCR5, CCR7, CXCR3, MCP-1), activation markers (CD62L, CD83, CD86, CD124, CD135, HLA-DR, IL-7R), transcription factors (Tbet, FOXP3), cytokines (IL-6, IL-8, IL-10, TGFβ, TNFα, IFNγ, GM-CSF) and metabolic markers (CD27, CD38, PD-1, PD-1 L, ADRP) served for a deep immune profiling of PBMCs (Supplementary Table 2).

We performed an unsupervised high-dimensional data analysis of CD45$^+$ cells using the t-distributed stochastic linear embedding (t-SNE) algorithm (Fig. 1e) and compared the frequency of cell subsets according to their expression levels of classical cell lineage markers, such as CD11b, CD3, CD4, CD8, CD14, CD19, and CD56, as well as functional, homing and activation markers, including CD86, CCR7, and HLA-DR (Fig. 1e, f). The expression levels of all markers in the different subpopulations were subsequently compared between healthy donors, CD patients and the AGLCD patient (Fig. 1g–l). To validate the degree of reproducibility of our mass cytometric data and to control for possible batch effects, we compared the expression of 16 overlapping immune markers included in both our mass cytometry antibody panels, revealing a high correlation between the two antibody panels, thus confirming the reliability of our findings (Supplementary Figs. 1 and 2).

In analogy to CD patients, we observed a reduced frequency of CD8a$^+$CCR7$^+$ (G1) T cells, as well as an increase in CD11b$^+$CD86$^+$ (G3) cells in the AGLCD patient when compared to healthy controls, highlighting a pre-activation of monocytes under inflammatory conditions and thus reflecting CD-induced changes in immune cell composition (Fig. 1e, f), which were further reflected

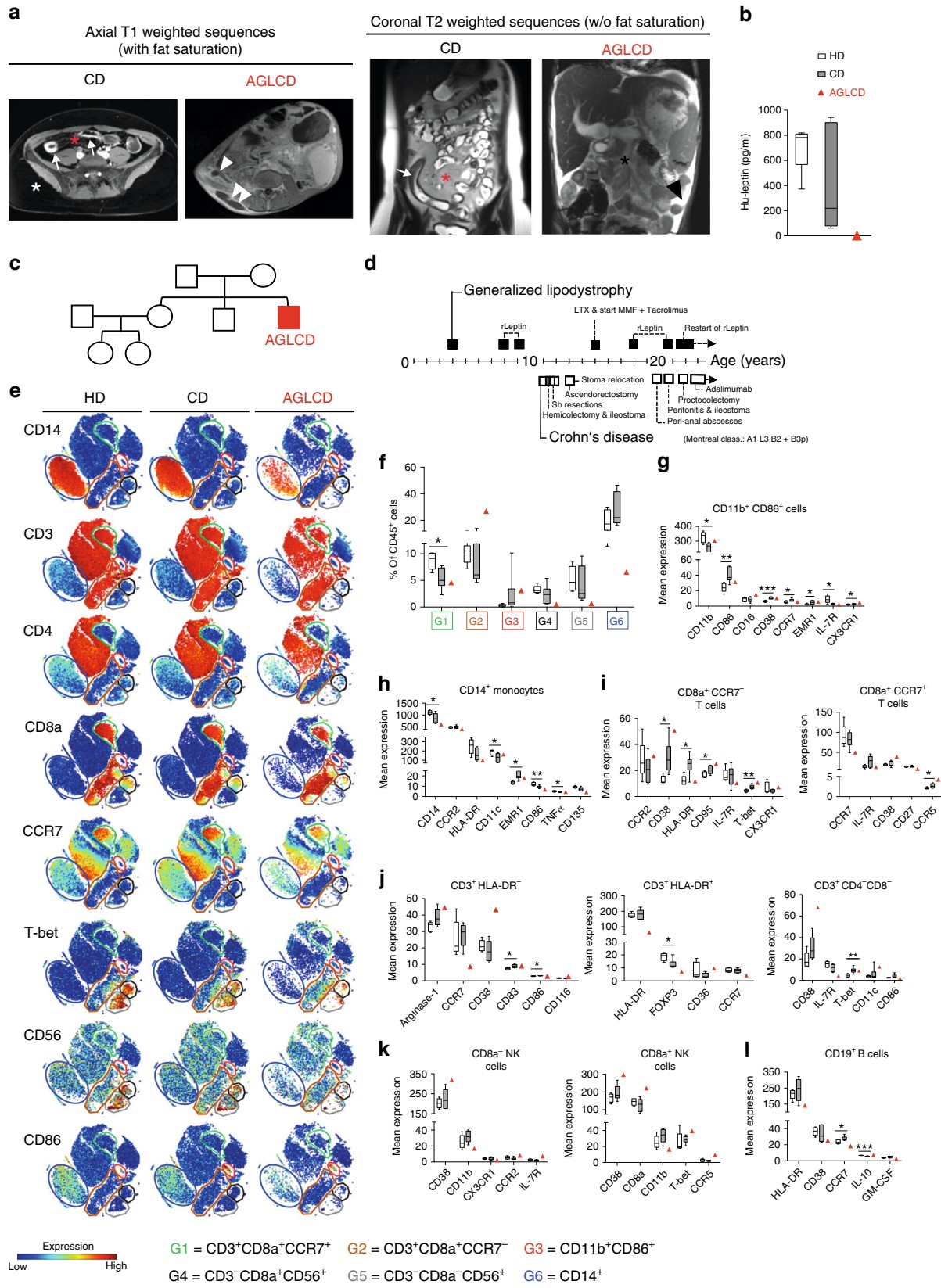

Expression
Low — High

G1 = CD3⁺CD8a⁺CCR7⁺    G2 = CD3⁺CD8a⁺CCR7⁻    G3 = CD11b⁺CD86⁺
G4 = CD3⁻CD8a⁺CD56⁺    G5 = CD3⁻CD8a⁻CD56⁺    G6 = CD14⁺

by comparable expression patterns of several differentiation and functional markers on CD11b⁺CD86⁺ cells and CD14⁺ monocytes of CD patients and the AGLCD patient (Fig. 1g, h).

In contrast, we found a severe reduction of CD14⁺ monocytes (G6) and of Tbet⁺CD56⁺CD8⁺ (G4) and Tbet⁺CD56⁺CD8⁻

NK cells (G5) in the AGLCD patient but not in CD patients or healthy donors (Fig. 1e, f), suggesting that these alterations are lipodystrophy-specific and at least partially caused by leptin deficiency as leptin receptor-deficient db/db mice also show decreased frequencies of NK cells[22].

**Fig. 1** 21-year-old male presenting with AGLCD and consecutively altered immune phenotype. The AGLCD patient was compared to Crohn' disease patients (CD) and healthy donors (HD). **a** MRI scans in two different sequences of a CD patient and the AGLCD patient, showing the complete lack of subcutaneous and visceral adipose tissue in the AGLCD patient. The white star indicates subcutaneous fat in the CD patient, the red star depicts mesenteric fat ("creeping fat") with injections of blood vessels wrapping inflamed intestinal segments (comb sign), the white arrows mark inflamed lesions in the CD patient. The single white arrowhead indicates a fistula in the AGLCD patient, the two white arrowheads depict a subcutaneous abscess, the single black star highlights the absence of mesenteric fat in the AGLCD patient, the black arrowhead marks free abdominal fluid. **b** Leptin serum concentrations assessed by ELISA in the AGLCD patient, CD patients ($n = 7$) and HD ($n = 5$) in biologically independent samples. **c** Family tree of the AGLCD patient. **d** Graphical summary of the AGLCD patient's clinical history. **e–l** Comparative immune profiling of PBMCs of the AGLCD patient, CD patients ($n = 6$) and HD ($n = 5$) by mass cytometry in biologically independent samples. **e** two-dimensional projections of single cell data generated by t-SNE of PBMCs. Desired subpopulations were gated (G1-G6). Heat colors of expression levels of selected markers have been scaled for each marker individually, while red denotes high and blue low expression. **f** Boxplots show the frequencies of different cell subsets (G1-G6). **g–l** Boxplots show mean expression levels (arbitrary unit) of selected markers in each cell subset. Boxes extend from the 25th to 75th percentiles. Whisker plots show the min (smallest) and max (largest) values. The line in the box denotes the median. *$P < 0.05$, **$P < 0.01$, ***$P < 0.001$, two-tailed unpaired t-test without correction for multiple comparison. The source data are provided as a Source Data file.

In comparison to CD patients and healthy donors, we furthermore detected a high expression of the activation marker CD38 on CD11b$^+$CD86$^+$, T and NK cells in the AGLCD patient (Fig. 1g, i–k), which has previously been linked to the development of lipodystrophy in HIV patients receiving anti-retroviral therapy[23] and to intestinal inflammation in human and mice[24] as DSS-induced colitis is attenuated in $CD38^{-/-}$ mice[25]. Of note, CD38 was significantly upregulated in CD8$^+$CCR7$^-$ T cells of CD patients underlining a potential disease-propagating role of CD38 in CD.

Furthermore, the AGLCD patient featured a reduced expression of CCR7 in his T and B cell compartment (Fig. 1i, j, l), suggesting an additional disturbance of lymphocyte trafficking as CCR7 has been implicated in the homing of lymphocytes to adipose tissue[26].

To functionally characterize the lymphocytes in the AGLCD patient, we stimulated PBMCs of the AGLCD patient, CD patients or healthy donors with ionomycin/PMA or LPS and determined TNFα and IFNγ production by flow cytometry (Supplementary Table 3 and Supplementary Fig. 3). As shown in Fig. 2a, b, we observed that the frequencies and mean fluorescence intensity (MFI) levels of TNFα and IFNγ producing CD4$^+$ and CD8$^+$ T, NK cells and CD14$^+$ monocytes were similar in cells isolated from the AGLCD patient when compared to CD patients and healthy donors. Likewise, the frequency of FOXP3 expressing CD4$^+$ T cells was comparable in the AGLCD patient and healthy donors (Supplementary Fig. 4).

**Impaired NK and T cell differentiation in the AGLCD patient.** Since we had found reduced frequencies of NK cells in the AGLCD patient (Fig. 1e, f) using mass cytometry, we further characterized his NK cell compartment by flow cytometry. Here, we found lower frequencies of NK cells expressing the cytotoxic molecules perforin and granzyme B, as well as a lower perforin expression in the AGLCD patient in comparison to healthy donors and CD patients. In contrast, CD patients displayed a significantly higher frequency of granzyme B$^+$ NK cells ($p < 0.001$) and an increased expression level of granzyme B ($p < 0.05$) when compared to healthy donors, whereas perforin expression was similar in both groups (Fig. 2c). In addition, we observed a disturbed pattern of CD56/CD16 expression in the AGLCD patient's NK cells compared to a healthy individual (Supplementary Fig. 4). These data indicate that the absence of fat tissue and therefore fat tissue-derived adipokines results in phenotypic and functional alterations of NK cells.

As the absent storage capacity of fat in adipocytes leads to hypertriglyceridemia in lipodystrophy patients[19] and it has been shown that intracellular accumulation of lipid droplets impairs the killing capacity of cytotoxic NK cells by metabolic reprogramming with subsequent downregulation of granzyme

B[27], we next measured the lipid droplet content in various immune cell subsets. As shown in Fig. 2d, the AGLCD patient displayed a higher deposition of lipid droplets in monocytes, NK cells and CD8$^+$ T cells when compared to CD patients and healthy donors. Interestingly, Greineisen et al. have recently described that lipid droplet accumulation decreases the capacity of lymphocytes to flux calcium[28], which is required to control the expansion of naïve CD8$^+$ T cells by orchestrating metabolic programming and inducing glycolysis[29]. In line with increased lipid droplets accumulation, we found an impaired calcium homeostasis in CD8$^+$ T cells (Fig. 2e), suggesting that the deranged lipid metabolism in the AGLCD patient contributes to the observed functional and metabolic immune cell dysfunctions. Please note that due to the very low frequency of NK cells in the AGLCD patient no reliable calcium measurements or functional killing assays could be obtained for NK cells. Glucose uptake, however, was not altered in PBMCs of the AGLCD patient (Supplementary Fig. 4). The observed changes in immune cell metabolism in the AGLCD patient were further reflected by the increased expression of CD38 in T and NK cells, as CD38, a multifunctional enzyme with nicotinamide adenine dinucleotide (NAD) nucleosidase activity[30], is not only a marker for activated T cells[31], but was also shown to regulate NAD$^+$ metabolism, oxidative phosphorylation and glutaminolysis in human CD4$^+$ T cells[32].

**rLeptin-induced TNFα production and immune cell alterations.** Since the AGLCD patient suffered from extreme insulin resistance due to lipodystrophy and consecutive leptin deficiency, we decided to restart rLeptin substitution, which had previously been initiated within the compassionate use program for rLeptin at the University of Leipzig but had been stopped by the patient months prior admission to our hospital due to incompliance.

The patient received daily subcutaneous injections with 2.5 mg rLeptin. After 4 days of rLeptin administration, a substantial serum concentration of leptin could be detected in the AGLCD patient (Fig. 3a). Consequently, rLeptin substitution reduced the required insulin concentrations of the patient ~10-fold and decreased the amount of lipid droplets in T, NK, and monocytic cells (Fig. 3b), which was also indicated by a decreased expression of ADRP, a lipid droplet binding protein (Fig. 3c). In line with this reduction in intracellular lipid droplet formation, the serum levels of triglycerides and total cholesterol also dropped and normalized under treatment with rLeptin (Supplementary Fig. 5). Strikingly, rLeptin substitution led to an increased pro-inflammatory activity of distinct immune cells, as reflected by an upregulation of perforin expression in NK and CD8$^+$ T cells (Fig. 3d), as well as an increased frequency and MFI of TNFα-expressing cells after ex vivo stimulation with ionomycin and

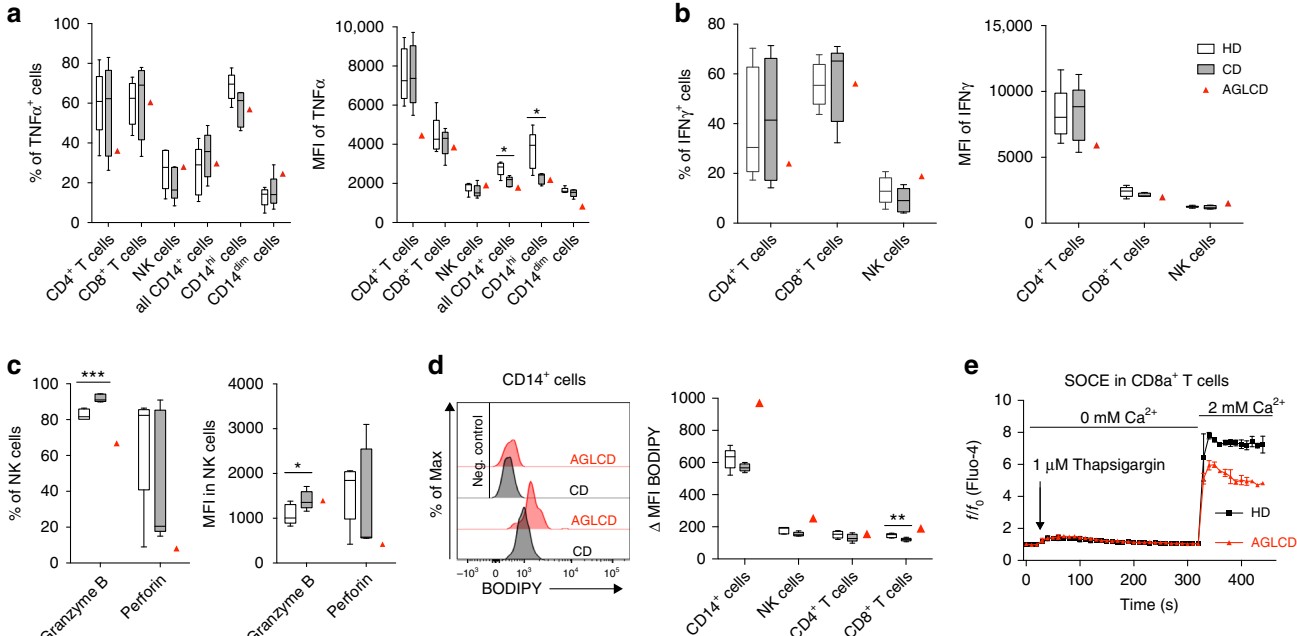

**Fig. 2** Functional and metabolic alterations of immune cells in the AGLCD patient. PBMCs of the AGLCD patient, Crohn's disease (CD) patients and healthy donors (HD) were compared by flow cytometry. **a–b** After ex vivo stimulation with PMA/ionomycin or LPS, the percentage of (**a**) TNFα-producing and (**b**) IFNγ-producing T, NK and monocytic cells were determined, as well as the respective mean fluorescence intensity (MFI) serving as a measure of the amount of cytokine production (CD: $n = 5$, HD: $n = 6$ for T and NK cells; CD: $n = 5$, HD: $n = 5$ for CD14$^+$ cells, biologically independent samples). **c** Unstimulated NK cells were analyzed for perforin and granzyme B expression regarding percentage of expressing cells and MFI (CD: $n = 5$, HD: $n = 5$, biologically independent samples). **d** Lipid droplet accumulation in immune cells was determined by BODIPY staining (CD: $n = 5$, HD: $n = 5$, biologically independent samples). **e** Store-operated Ca$^{2+}$ entry (SOCE) was measured in CD8$^+$ T cells comparing AGLCD patient with HD ($n = 1$, both in technical duplicates). Boxes extend from the 25th to 75th percentiles. Whisker plots show the min (smallest) and max (largest) values. The line in the box denotes the median. Error bars on the SOCE plot represent the standard deviation (SD). *$P < 0.05$, **$P < 0.01$, ***$P < 0.001$, two-tailed unpaired $t$-test without correction for multiple comparison. The source data are provided as a Source Data file.

PMA (Fig. 3e). In addition to these functional changes, we also detected an expansion of activated CD11b$^+$CD86$^+$ cells (G3) seven days after rLeptin application, whereas reduced numbers of CD8a$^+$CCR7$^-$ T cells (G2), CD19$^+$ B cells and CD8a$^+$ NK cells (G4) were found after treatment with rLeptin (Fig. 3f, g).

In our view, the observed phenotypic and functional changes of NK cells in response to rLeptin treatment, including a change in CD56 expression pattern (Supplementary Fig. 6), granzyme B and perforin expression, as well as cytokine production, demonstrate that leptin is an important regulator of NK cell differentiation and function. Accordingly, leptin receptor-deficient mice show impaired NK cell development and function[22], whereas leptin administration increases the proliferation, cytotoxic capacity and perforin expression in the human NK cell line YT[33]. However, as indicated by the NK cell numbers remaining low even after rLeptin substitution, other fat tissue-derived factors such as adipokines and metabolites likely contribute to proper NK cell development and might be missing in the fat-deficient AGLCD patient.

Deep immune profiling of PBMCs by mass cytometry showed additional rLeptin-associated pro-inflammatory phenotypes in various immune cell subpopulations in the AGLCD patient (Fig. 3c and Supplementary Fig. 6). For example, in CD11b$^+$CD86$^+$, T and NK cells, we found a reduced arginase-1 and increased CD86 expression following rLeptin treatment (Fig. 3c). Arginase-1 has been proposed as a marker for alternatively-activated macrophages[34] and is an important intrinsic regulator of amino acid metabolism and glycolysis in innate lymphoid cells type 2[35]. Furthermore, we observed decreased CD38 expression in T, NK and CD11b$^+$CD86$^+$ cells following rLeptin administration, which

supports the assumption of leptin-dependent regulations of NAD$^+$ metabolism in these cells with downstream effects on glutaminolysis and energy metabolism[30,32]. In addition, we found increased serum concentrations of several pro-inflammatory, monocyte-derived cytokines including G-CSF, MIG and MIP-1β upon rLeptin application (Fig. 3h), suggesting that rLeptin substitution results in increased activation of myeloid cells.

Metabolic reprogramming of immune cells by adipocytes and adipokines was shown to trigger specific differentiation programs, thereby determining their function[36,37]. Thus, pro-inflammatory macrophages highly depend on glycolysis for their energy homeostasis, whereas fatty acid oxidation is a feature of macrophages with anti-inflammatory properties[38]. Accordingly, when we differentiated monocyte-derived macrophages of a healthy donor in the presence of serum obtained from the AGLCD patient after rLeptin substitution, cells showed a reduced mitochondrial respiration when compared to macrophages that received AGLCD serum without rLeptin (Fig. 3i, j), whereas their extracellular acidification rates (ECAR) were comparable (Supplementary Fig. 7). Likewise, in vitro expanded CD8$^+$ T cells featured an altered oxygen consumption and extracellular acidification rate after incubation with AGLCD serum substituted with rLeptin when compared to controls receiving serum of the AGLCD patient without rLeptin (Supplementary Figs. 8 and 9), suggesting that leptin directly influences the bioenergetics of immune cells.

Since intestinal epithelial cells express the leptin receptor[39] and direct effects of leptin on epithelial cells have been reported[2], we wanted to investigate if leptin might influence epithelial barrier homeostasis in intestinal inflammation by looking at the wound

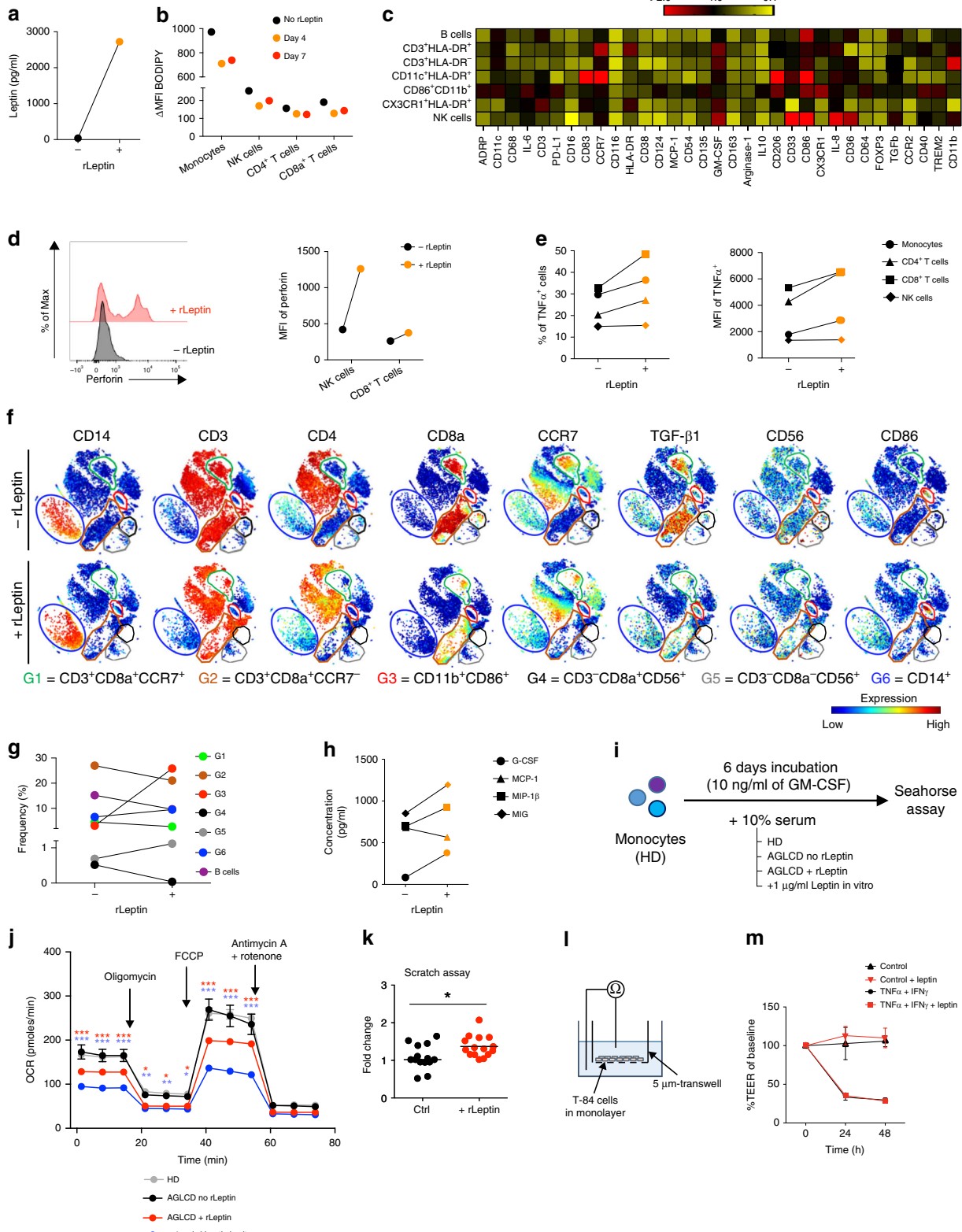

G1 = CD3⁺CD8a⁺CCR7⁺    G2 = CD3⁺CD8a⁺CCR7⁻    G3 = CD11b⁺CD86⁺    G4 = CD3⁻CD8a⁺CD56⁺    G5 = CD3⁻CD8a⁻CD56⁺    G6 = CD14⁺

healing capacity and epithelial resistance of human intestinal epithelial cells in vitro. As shown in Fig. 3k leptin administration improved wound closure in vitro in scratch assays with T84 cells, without altering epithelial resistance (Fig. 3l, m).

**Intestinal inflammation under rLeptin substitution.** The high disease activity of CD observed in our AGLCD patient during rLeptin substitution and previously developed structural damage,

in particular fistula development, resulted in an abscess that ultimately led to ileocolonic resection and terminal ileostomy (Figs. 1d and 4a). Severe inflammation was detected in all resected intestinal segments (Fig. 4b). Results from immunohistochemistry (Supplementary Table 4) showed a massive infiltration of the intestinal mucosa with TNFα-expressing cells within the lamina propria of the AGLCD patient when compared to CD patients (Fig. 4c). In line with our mass cytometry data, we

**Fig. 3** Leptin induces pro-inflammatory immune responses in the AGLCD patient and improves wound healing in vitro. **a** Leptin serum concentrations of the AGLCD patient before (−) and after (+) 4 days of rLeptin substitution. **b–g** Flow and mass cytometric analysis of the effects of rLeptin treatment on PBMCs in the AGLCD patient after 4 and/or 7 days. **b** Lipid droplet content assessed by BODIPY staining. **c** Heat map displaying the relative changes in mean expression (compared to before rLeptin substitution) of various functional markers in different cell subsets (mass cytometry antibody panel B in Supplementary Table 2) (red, fold change above 2 (increased expression); yellow, fold change = 0.1 (decreased expression)). **d** Perforin expression measured as mean fluorescence intensity (MFI). **e** TNFα-expressing cells and the respective MFI. **f** Two-dimensional projections of single cell data generated by t-SNE of mass cytometry data (antibody panel A in Supplementary Table 2) show the effects on (**g**) frequencies of different cell subsets. Heat colors of expression levels of selected markers on t-SNE maps have been scaled for each marker individually (red, high expression; blue, low expression). **h** Serum levels of different immune cell-derived factors measured by CBA **i–j** Oxygen consumption rate (OCR) assessed by Seahorse analyses in monocyte-derived macrophages of a healthy donor differentiated in the presence of serum from the AGLCD patient before ("leptin-free") and after in vitro or in vivo leptin/rLeptin substitution (performed in at least triplicates, error bars show ± SEM, two-way ANOVA with post-tests and Holm-Sidak correction). The corresponding extracellular acidification rates (ECAR) are reported in Supplementary Fig. 7. **k** Scratch assay with human T84 intestinal epithelial cells in the presence of leptin to assess in vitro wound healing. **l–m** Effects of leptin on the transepithelial electrical resistance (TEER) across a monolayer of T84 cells after challenge with TNFα and IFNγ to induce leakage (error bars show SD). *$P < 0.05$, **$P < 0.01$, ***$P < 0.001$, two-tailed unpaired t-tests (where applicable). The source data are provided as a Source Data file.

furthermore observed a higher infiltration with activated CD86+ cells in the AGLCD patient (Fig. 4d, e), indicating that the AGLCD patient had a TNFα-dependent inflammatory activity in his intestine at the time of surgery. Considering our data on the effects of rLeptin treatment on the AGLCD patient's PBMCs (Fig. 3), we therefore hypothesized that rLeptin treatment induced a TNFα-driven pro-inflammatory immune response in the intestine, triggering a severe exacerbation of CD.

**Stable remission of the AGLCD patient under TNFα−blockade.** Following surgery, the patient continued rLeptin substitution to improve insulin sensitivity and subsequently again developed mucosal inflammation beginning at the terminal ileostomy, further supporting the concept of rLeptin-driven intestinal inflammation. Due to the increased production of TNFα that we had noted upon rLeptin substitution earlier, we decided to initiate anti-TNFα therapy with adalimumab, resulting in stable clinical and endoscopic remission after 6 months of therapy (Fig. 4f). Moreover, TNFα blockade not only inhibited TNFα and IFNγ production in CD4+ and CD8+ T cells but also increased the frequency of FOXP3+CD4+ T cells in the AGLCD patient as determined by flow cytometry (Fig. 4g–i).

## Discussion

In summary, our results shed light on the complex regulatory role of fat tissue in intestinal inflammation, as well as on the immune stimulatory effects of rLeptin on human immune cell composition, function and metabolism (as summarized in Supplementary Fig. 10). Farooqi et al. have previously demonstrated that leptin substitution promotes T cell proliferation and cytokine production in three patients with congenital leptin deficiency. However, in the absence of a concomitant inflammatory disease, no inflammatory response was detected in vivo[16]. Likewise, Oral et al. have observed increased TNFα production in PBMCs in a cohort of 10 patients with generalized lipodystrophy after four months of treatment with rLeptin without noticing an induction or exacerbation of comorbid inflammatory diseases and consequently no autoimmune complications were reported[40]. Our results now indicate that, in the setting of a pre-existing inflammatory condition, leptin therapy fuels inflammation and increases disease activity in autoimmunity via the induction of TNFα-producing cells and by metabolically priming immune cells towards a pro-inflammatory phenotype. Consistently, Javor et al. have previously speculated that rLeptin substitution might have contributed to the deterioration of membranoproliferative glomerulonephritis by inducing autoimmunity in two patients with generalized lipodystrophy in a study investigating the effects of

rLeptin substitution on the renal function of lipodystrophic patients[41]. Of note, the AGLCD patient received immunosuppressive medication throughout our study, which is likely to have affected his immune cell composition. However, the medication was not modified during the course of our study and the described pro-inflammatory effects of rLeptin were observed despite immunosuppression with mycophenolate mofetil (MMF) and tacrolimus. Furthermore, we cannot fully exclude that treatment with MMF contributed to intestinal inflammation in the AGLCD patient as MMF-therapy has been associated with chronic diarrhea, which can be reversed by anti-TNFα therapy[42,43]. However, we believe that the histologic presence of granulomas (Supplementary Fig. 11), the fistulizing nature of inflammation (Fig. 1a), as well as the occurrence of intestinal autoimmunity prior to the start of MMF strongly argue against MMF-driven intestinal inflammation in the AGLCD patient, since none of these features can be observed in MMF-induced colitis[44]. Even if we do not consider MMF to trigger intestinal inflammation in the AGLCD patient, anti-TNF therapy would have been beneficial both for treating leptin-induced inflammation, as well as potential side effects caused by MMF.

Since our findings are based on observations in a single patient with AGLCD with an extremely complicated clinical course (Fig. 1d), the generalizability of our findings might be limited and it would be important to further validate our findings in additional patients with acquired generalized lipodystrophy and concomitant autoimmune disease, which is unfortunately difficult to perform due to the extreme rarity of these patients and the singularity of the AGLCD patient. However, we are convinced that our results still provide valuable insights regarding the immune-stimulatory potential of leptin in human intestinal inflammation, which are supported by observations in mice as leptin-deficient ob/ob mice are protected from DSS-induced colitis[6] and the pharmacologic inhibition of the leptin receptor attenuates disease severity in Il10−/− mouse models of colitis[7]. In our opinion, leptin thereby does not directly trigger inflammation (otherwise rLeptin replacement should cause autoimmunity in all forms of lipodystrophy, which is not the case), but is instead enhancing autoimmunity by facilitating the production of pro-inflammatory cytokines such as TNFα and by regulating immune cell differentiation and cellular expansion of auto-reactive lymphocytes. In addition, our observations also argue in favor of a broader role of leptin for proper immune cell function as leptin deficiency is associated with a decreased and impaired NK compartment both in the AGLCD patient and leptin receptor-deficient db/db mice[22], suggesting that leptin-deficiency might be considered as a cause of functional immune deficiency. Accordingly, patients with malnutrition and consecutive low levels of leptin suffer from an increased susceptibility for severe infections

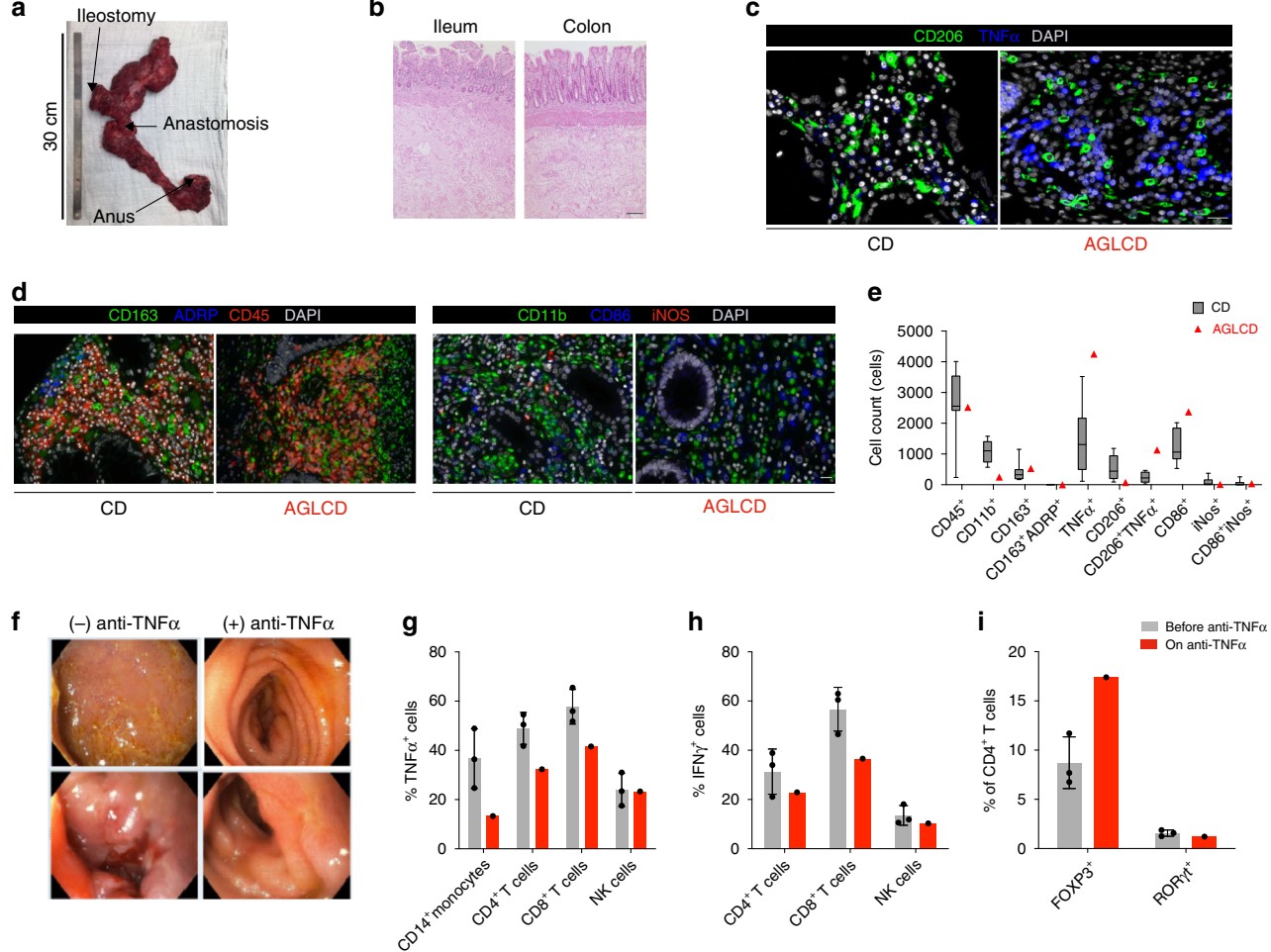

**Fig. 4** TNFα-driven intestinal inflammation during rLeptin treatment is reversed by anti-TNFα therapy. **a–e** Sustained inflammation under rLeptin substitution and previously acquired structural damage (stenosis and fistula-induced abscess) made proctocolectomy and ileal resection necessary 16 days after initiation of rLeptin therapy. The specimen (**a**) was analyzed histologically by H&E staining (scale bar depicts 100 μm) (**b**), showing severe inflammation, and by immunohistochemistry (IHC; **c–e**). **c** Microscopic pictures of IHC staining for TNFα and CD206 in gut tissue from the AGLCD patient and a Crohn's disease (CD) patient (The scale bar depicts 20 μm, images were recorded using an AxioImager Z1 from Zeiss). **d** Additional pictures of IHC staining for different immune cell markers (recorded with the Vectra3 system from PerkinElmer, the scale bar displays 20 μm) and (**e**) quantification of cells staining positive for respective markers per 10 high power fields (For CD45, CD163 and ADRP stainings a total of 11 tissue samples from $n = 7$ biologically independent CD control patients was compared to the AGLCD patient ($n = 1$); for TNFα and CD206 stainings a total of 9 samples of $n = 6$ independent CD patients was obtained and compared to the AGLCD patient ($n = 1$); for CD11b, CD86 and iNOS stainings a total of 6 samples from $n = 3$ biologically independent CD patients was compared to the AGLCD patient ($n = 1$); in case that several samples were analyzed from the same patients, tissues were derived from different anatomical locations, blinded analysis). The box extends from the 25th to 75th percentiles. Whisker plots show the min (smallest) and max (largest) values. The line in the box denotes the median. **f–i** Following surgery, rLeptin treatment was continued and the patient presented with new inflammation beginning at the terminal ileostomy. The clinical decision was made to initiate anti-TNFα therapy while continuing rLeptin substitution, resulting in clinical and endoscopic remission of Crohn's disease. **f** Comparison of pictures taken during endoscopy before starting anti-TNFα therapy and six months later. **g–i** Flow cytometric analysis of the effects of anti-TNFα therapy on (**g**) TNFα and (**h**) IFNγ production after ex vivo stimulation, as well as (**I**) the expression of the transcription factors FOXP3 and RORγt ("before anti-TNFα" combines data from three different time points). Error bars on column charts display the standard deviation (SD). The source data are provided as a Source Data file.

including leishmaniosis and amiobiasis due to impaired T and NK cell functions[45].

We are aware, that our study does not allow to discriminate between direct and indirect effects of rLeptin substitution on immune cells as a variety of cofounding metabolic factors could contribute to the observed immune stimulatory function of leptin in the AGLCD patient including the leptin-mediated regulation of appetite, blood lipids, glucose and insulin sensitivity. However, Reis and colleagues could recently unambiguously show, that leptin has direct primary effects on the function and differentiation of lymphocytes in intestinal autoimmunity as leptin-receptor deficient CD4+ T cells from *Lepr^{fl/fl}-CD4-Cre* mice fail to

induce intestinal inflammation in transfer models of colitis[46]. Accordingly, our group has observed a significantly delayed onset of colitis in *scid* mice after the transfer of CD4+ T cells from Leptin-receptor deficient *db/db* mice due to a defective production of inflammation-promoting cytokines including IFNγ[11] further supporting the pro-inflammatory function of leptin in inflammatory bowel disease.

Our results furthermore help to better understand the complex role of mesenteric fat in intestinal inflammation, as it is difficult to study in animal models. Remarkably, Paul et al. have previously described that leptin is elevated in creeping fat of Crohn's disease patients[47]. Accordingly, CD patients with a high burden

of creeping fat also display elevated levels of leptin in the serum and the amount of visceral fat is directly correlated with a higher disease activity[48]. In line with this observation that creeping fat serves as a local source for leptin and other pro-inflammatory cytokines, Coffey and colleagues have recently registered in a small retrospective study that a radical resection of the mesentery and creeping fat in CD patients increases the recurrence free survival of patients[49]. Our group has previously observed in mice, that intestinal inflammation and subsequent bacterial translocation into the mesenteric fat induces the local production of leptin by adipocytes in a TLR-dependent manner in DSS treated wild-type mice but not in $Myd88^{-/-}$ mice[50]. Given the high infiltrations of creeping fat with lymphocytes such as $CD8^+$ T cells and monocytes[51], it is plausible to argue that local leptin production affects the differentiation, as well as the functionality of fat residing lymphocytes, especially in regard of the observation that T cells express leptin receptors and that leptin receptor-deficient T cells fail to induce colitis in mice[11,46]. However, additional studies are required to decipher the immune-modulatory function of creeping fat in the pathogenesis of Crohn's disease which is likely to consist of cumulative effects of additional adipokines and metabolites that are secreted by creeping fat.

Taken together, we believe that our findings might have important implications for the treatment of patients receiving rLeptin substitution. Since acquired generalized lipodystrophy is associated with autoimmune and inflammatory diseases in about 25% of cases[19], there may be more lipodystrophy patients in whom rLeptin therapy is accompanied by an increased activity of a concomitant inflammatory disease. Therefore, our study provides first evidence that in such cases anti-TNFα therapy should be considered to control autoimmunity while maintaining rLeptin substitution for the metabolic control of lipodystrophy.

## Methods

**Ethical regulations**. Written informed consent was obtained from all patients and healthy volunteers and included the consent to publish clinical information potentially identifying individuals. All experiments were approved by the institutional review board of the Charité–Universitätsmedizin Berlin and conducted accordingly. The authors complied to all relevant ethical regulations for research involving human participants.

**Enzyme-linked immunosorbent assay (ELISA)**. Serum was obtained from peripheral blood of the AGLCD patient before and after rLeptin (metreleptin from Aegerion Pharmaceuticals) treatment, as well as from 7 CD patients and 5 healthy controls. Serum levels of leptin were determined using the Human Leptin DuoSet ELISA Kit (R&D Systems).

**Isolation of peripheral blood mononuclear cells**. Heparinized blood was obtained from the AGLCD patient at different time points before and after leptin reconstitution. Blood of healthy donors, and CD patients served as controls. PBMCs were isolated by density gradient centrifugation using Biocoll (Merck). Cells were either frozen in RPMI 1640 (ThermoFisher Scientific) substituted with 20% fetal bovine serum (FBS; Sigma) and 10% dimethyl sulfoxide (Sigma) and stored in liquid nitrogen or used directly. For mass cytometry, cells were fixed and frozen using SmartTube buffer and subsequently stored at −80 °C according to our previously published protocol[21].

**Intracellular barcoding for mass cytometry**. SmartTube buffer-fixed PBMCs of the AGLCD patient before and after rLeptin substitution, as well as cells of 6 CD patients and 5 HD were thawed and subsequently stained with premade combinations of six different palladium isotopes: $^{102}Pd$, $^{104}Pd$, $^{105}Pd$, $^{106}Pd$, $^{108}Pd$, and $^{110}Pd$ (Cell-ID 20-plex Pd Barcoding Kit, Fluidigm). After 30 min staining at room temperature (RT), individual samples were washed twice with cell staining buffer (0.5% bovine serum albumin in PBS, containing 2 mM EDTA). Samples were pooled together, washed and further stained with antibodies. Anti-human antibodies (Supplementary Table 2) were purchased either pre-conjugated to metal isotopes (Fluidigm) or from commercial suppliers in purified form and conjugated in house using the MaxPar X8 kit (Fluidigm) according to the manufacturer's protocol.

**Surface and intracellular staining for mass cytometry**. After cell barcoding, washing and pelleting, the combined samples were re-suspended in 100 μl of antibody cocktail against surface markers (Supplementary Table 2) and incubated for 30 min at 4 °C. Then, cells were washed twice with cell staining buffer. For intracellular staining, the stained (non-stimulated) cells were then incubated in fixation/permeabilization buffer (Fix/Perm Buffer, eBioscience) for 60 min at 4 °C. Cells were then washed twice with permeabilization buffer (eBioscience). The samples were then stained with antibody cocktails against intracellular molecules (Supplementary Table 2) in permeabilization buffer for 1 h at 4 °C. Cells were subsequently washed twice with permeabilization buffer and incubated overnight in 2% methanol-free formaldehyde solution (ThermoFisher). Fixed cells were then washed and re-suspended in 1 ml iridium intercalator solution (Fluidigm) for 1 h at RT. Next, the samples were washed twice with cell staining buffer and then twice with ddH$_2$O (Fluidigm). Cells were pelleted and kept at 4 °C until CyTOF measurement.

**CyTOF measurement**. Cells were analyzed using a CyTOF2 upgraded to Helios specifications, with software version 6.5.236. The instrument was tuned according to the manufactures instructions with tuning solution (Fluidigm) and measurement of EQ four element calibration beads (Fluidigm) containing $^{140/142}Ce$, $^{151/153}Eu$, $^{165}Ho$, and $^{175/176}Lu$ served as a quality control for sensitivity and recovery. Directly prior to analysis cells were re-suspended in ddH$_2$O, filtered (20 μm Celltrix, Sysmex), counted and adjusted to $3–5 \times 10^5$ cells/ml. EQ four element calibration beads were added at a final concentration of 1:10 of the sample volume to be able to normalize the data to compensate for signal drift and day-to-day changes in instrument sensitivity. Samples were acquired with a flow rate of 300–400 events/s. Lower convolution threshold was set to 400, with noise reduction mode on and cell definition parameters set at event duration of 10–150. The resulting flow cytometry standard (FCS) files were normalized and randomized using the CyTOF software's internal FCS-Processing module on the non-randomized ('original') data. Settings were used according to the default settings in the software with time interval normalization (100 s/minimum of 50 beads) and passport version 2. Intervals with less than 50 beads per 100 s were excluded from the resulting fcs-file. Cytobank (www.cytobank.org) was used for initial manual gating on live single cells, de-barcoding and viSNE to generate t-SNE maps. FCS files containing the t-SNE embedding as additional two parameters were exported from Cytobank for downstream exploratory and statistical analyses using GraphPad Prism 6.

**Flow cytometry**. For the assessment of cytokine production, PBMCs of the AGLCD patient before and after rLeptin substitution, as well as PBMCs from 5 CD patients and 6 HD were thawed and cultured for 4 h in RPMI 1640 supplemented with 10% fetal bovine serum and containing 20 ng/ml phorbol 12-myristate 13-acetate (PMA; Sigma-Aldrich) and 1 μg/ml ionomycin (Sigma-Aldrich) or 100 ng/ml lipopolysaccharide (Sigma-Aldrich). Brefeldin A (10 μg/ml; Sigma-Aldrich) was added for 2 h prior to harvesting. Cells were then stained for viability using a fixable dye (LIVE/DEAD™ aqua; ThermoFisher Scientific or Zombie Violet™; BioLegend). Subsequently, lineage markers (Supplementary Table 3) were stained and the Foxp3 Staining Set (ThermoFisher Scientific) was applied for fixation and intracellular and intranuclear staining according to the manufacturer's protocol (Supplementary Table 3). Samples were measured using a Canto II flow cytometer (BD Bioscience). Data were analyzed using the FlowJo software package V10.1 (FlowJo, LLC).

**Lipid droplet formation**. Thawed PBMCs were incubated for 30 min with 0.2 μg/ml BODIPY dye (ThermoFisher Scientific) and subsequently stained and analyzed by flow cytometry, as described above. The difference of mean fluorescence intensity (MFI) between BODIPY stained and all-but-BODIPY-stained samples (ΔMFI) served as a measure of lipid droplet content.

**Glucose influx**. To assess glucose uptake in immune cells, PBMCs of the AGLCD patient or a HD were glucose starved for 30 min in PBS. Subsequently, 100 μM of the fluorescent glucose analog 2-NBDG (ThermoFisher Scientific) was added to the cells and the MFI of 2-NBDG was assessed in different immune cell subsets using flow cytometry. The ΔMFI calculates as described for BODIPY experiments.

**Calcium measurements**. For calcium measurements, thawed PBMCs of a HD and the AGLCD patient were loaded for 30 min with the calcium sensing dye Fluo-4 AM (2 μg/ml; ThermoFisher Scientific) and subsequently stained for lineage markers. Calcium influx was measured by flow cytometry as previously described[52]. The experiment was performed in technical duplicates. Data were analyzed with FlowJo V8.8.7.

**Cytometric bead array**. Concentrations of G-CSF, MIG, MCP-1 and MIP-1β were measured in serum of the AGLCD patient before and 4 days after rLeptin substitution by using the FlowCytomix multi-plex kit from eBioscience according to the manufacturer's protocol. Data were analyzed using the FCAP Array™ software V3.0 from BD Biosciences.

**Seahorse analysis**. Monocytes, CD4+ or CD8+ T cells were isolated from PBMCs of healthy donors using bead-based positive selection kits for CD14+, CD4+, or CD8+ cells, respectively (Miltenyi Biotec). Monocytes were polarized into macrophages by adding 10 ng/ml GM-CSF (PeproTech) to the medium, while T cells were activated by stimulation with plate bound anti-CD3 (from eBioscience, clone OKT3) and anti-CD28 (from BD Bioscience, clone CD28.2) antibodies for 3 days and subsequently expanded for 4 days in media containing 20 ng/ml IL-2 (PeproTech). Throughout all experiments, RPMI 1640 media was supplemented either with 10% serum obtained from the AGLCD patient before or after rLeptin reconstitution (±rLeptin), with AGLCD serum (−rLeptin) that had been supplemented with 1 μg/ml leptin in vitro or with serum from healthy donors (HD). After 6 or 7 days of culture, the oxygen consumption rate and the extracellular acidification rate was assessed in at least triplicates on a 96-well Seahorse FX plate reader using the Cell Mito Stress Test Kit (Agilent).

**Histopathology**. Paraffin blocks of intestinal tissues derived from the AGLCD patient and 6–11 samples from different anatomical sites of 3 to 7 CD patients were cut and stained with hematoxylin and eosin (H&E) or by immunohistochemistry using antibodies listed in Supplementary Table 4. For antibody detection, the Opal 4-Color Manual IHC Kit (PerkinElmer) was used. Multispectral images were acquired using a Vectra® 3 imaging system (PerkinElmer). Positive cells were quantified in 10 high power fields (field of vision in ×400 original magnification) by inForm software (PerkinElmer). All evaluations were performed in a blinded manner. For the acquisition of immunofluorescence images depicted in Fig. 4d an AxioImager Z1 (Zeiss) was used.

**Scratch assay**. Human T84 colon epithelial cells were obtained from ATCC and cultured in 24-well plates and scratches were induced using small pipet tips. 1 μg/ml recombinant human leptin or vehicle was added (Peprotech). Pictures of each scratch were taken at the start and end (8 days or until first scratch was closed) using the Axiocam 105-color camera on a Zeiss Primovert microscope. Scratch areas were determined using the MiToBo Scratch Assay Analyzer (ImageJ) and changes in scratch area (Δarea) were normalized to the mean Δarea of the control group.

**Transepithelial electric resistance measurements**. Human T84 cells were cultured as a monolayer on transwell inserts (5-μm pores from Sigma) placed in a 24-well plate containing DMEM high glucose medium (4.5 g/l glucose; from ThermoFisher Scientific), 10% FBS and 1% penicillin/streptomycin. At baseline, 1 μg/ml leptin or vehicle (PBS) was added to cells. To induce epithelial leakage, 2 ng/ml recombinant human TNFα and 2 ng/ml IFNγ (both from Peprotech) were added to the medium. Electrical resistance was measured between an electrode in the well and an electrode in the transwell at baseline, after 24 and 48 h, respectively. Transepithelial electric resistance (TEER) was determined as the measured electrical resistance minus the resistance in uncultured condition (130 Ω) multiplied with the area of the transwell (0.6 cm$^2$). Results were normalized to the TEER at baseline of that respective well.

**Statistical analysis and graphs**. Statistical analysis and graphic data representation were done using GraphPad Prism version 7.00. Where not indicated otherwise, two-tailed unpaired $t$-tests without correction for multiple comparison were used. For Seahorse analysis, a two-way ANOVA with post-tests comparing against "AGLCD (-rLeptin)" as control with the Holm-Sidak correction for multiple comparison was applied. Results were considered statistically significant if $p < 0.05$.

**Exome sequencing**. Blood-derived DNA was extracted from the patient, as well as his healthy mother, sister and brother. Exome enrichment for these four samples was performed using the IDT xGen Exome Research Panel v 1.0 and $2 \times 75$ bp paired-end sequencing was carried out on an Illumina HiSeq 3000. The reads were mapped against the human reference genome build hg19 using BWA[53], sorted, converted to bam format and indexed with SAMtools[54], followed by the removal of PCR duplicates, local realignment around InDels and base quality score recalibration with the GATK[55] according to their best practice recommendations followed by variant calling and variant quality score recalibration. Variant annotation and filtering were performed using Alissa Interpret (Agilent).

**Reporting summary**. Further information on research design is available in the Nature Research Reporting Summary linked to this article.

## Data availability

The source data underlying Figs. 1b, 1f–l, 2a–e, 3a–e, 3g–h, 3j–k, 3m, 4e, 4g–i and Supplementary Figs. 1A–E, 2A-C, 4A, 4C-E, 5, 6A-C, 6E, 7, 8 and 9A are provided as a Source Data file. All mass and flow cytometric data sets, as well as exome sequencing results generated and analyzed during this study are available from the corresponding authors on reasonable request that does not include confidential patient information as the AGLCD patient and his relatives did not consent to a deposition of their personal data on a public repository.

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

## Acknowledgements

This work was funded by the German Research Foundation (We 5303/3-1 to C.W., Si 749/10-1 to B.S., SFB-TRR 241 B01 and INF to A.A.K., B.S. and C.W.), and the German-Israeli Foundation for Scientific Research and Development to B.S. and R.G., J.Z. was funded by a scholarship from the Deutsche Gesellschaft für Innere Medizin, C.W. received funding by the Clinician Scientist Program of the Berlin Institute of Health. C.B. and J.P. were supported by the German Research Foundation (SFB-TRR167, B05 and B07). J.P. received additional funding from the Berlin Institute of Health (CRG2aSP6) and the UK DRI (Momentum Award). A.A.K., B.S., and P.A. were supported by the German Research Foundation (SFB-1340, B06 and B07), F.T., B.P., B.F., M.F., A.F. were supported by the DFG Cluster of Excellence "Inflammation at Interfaces" and the Bundesministerium für Bildung und Forschung (E:med/SysInflame,012 × 1306 F). We would like to acknowledge the assistance of the BIH Cytometry Core Facility (BIH and Charité–Universitätsmedizin Berlin, Germany).

## Author contributions

J.F.Z., C.B., M.L., C.Y., H.W., I.F., Y.R.S., A.A.K., D.K., R.G., F.T., B.L. performed experiments, J.F.Z., C.B, M.L., C.Y., H.W., A.A.K., R.G., F.T., B.F., B.L., B.S., and C.W. designed and analyzed experiments, J.F.Z., A.K.S., M.E.K., I.F., Y.R.S., I.A., C.Boj., P.A., A.A.K., K.M., M.S., B.S., C.W. contributed to sample acquisition and the clinical management of the AGLCD patient, J.P., I.A., P.A., R.G., B.F., M.F., and A.F. helped with data interpretation, J.F.Z., C.B., B.S., and C.W. wrote the paper.

## Competing interests

K.M. consulted for Aegerion Pharmaceuticals. B.S. has served as consultant for Abbvie, Boehringer, Celgene, Falk, Janssen, Lilly, Pfizer, Prometheus, Takeda and received speaker's fees from Abbvie, CED Service GmbH, Falk, Ferring, Janssen, Novartis, Takeda (B.S. served as representative of the Charité). The remaining authors declare no competing interests.
