## [Peer Review File · Nature Communications]

REVIEWERS' COMMENTS:

Reviewer #1 (Remarks to the Author):

The Authors have satisfactorily addressed all the issues raised.

Reviewer #4 (Remarks to the Author):

The revisions in the manuscript and supplemental figures address the concerns raised in the review.

Reviewer #5 (Remarks to the Author):

The authors have properly addressed raised comments by Reviewer 2 and thereby substantially improved this work.

Reviewer #6 (Remarks to the Author):

I comment, as requested, on the likely general importance of this report, especially from a metabolic and lipodystrophy perspective, following on from previous reviewer 3, with whose views I essentially concur. The is an interesting and detailed opportunistic study of the immunological consequences of recombinant human leptin therapy used in the context of acquired generalised lipodystrophy, previous liver transplantation, and Crohn's disease, which I have not come across associated with LD before in either published or unpublished cases.

In response to previous reviews referencing of prior literature has been improved, and the paper generally is well written and clear.

The cellular and immunological work up is quite impressive given a single case and limited material, but the major problem is, of course, that this is a single, very complicated case, leaving some doubt about generalisability of findings.

The authors do a good job of placing their findings in the context of prior studies of immunological effects of leptin, but therein lies another problem. It has long been suggested that leptin deficiency is somewhat immunosuppressant, and that leptin reconstitution may reactivate quiescent autoimmune disease, but this has been backed only by a handful of clinical anecdotes, and by evaluation of effects of leptin in people without overt autoimmune disease. This means that the current case is of value, but equally that the information provided is also rather incremental.

Possible mechanisms underlying the observations are rationalised with reference to other studies, for example in db/db mice, but this study alone cannot discriminate direct and indirect effects of leptin. Leptin in LD reduces appetite, improves blood lipid, glucose and insulin levels, and in principle any of these may contribute to changes in immune cell function. Without controls such as those with severe insulin resistance who are leptin replete, those with poorly controlled insulin deficient diabetes, or those with absent leptin levels but no diabetes and insulin resistance (and these can be found) indirect effects of leptin cannot be ruled out.

Another question is the wider importance of a possible role of leptin in immune function in Crohn's. In metabolic control leptin appears to have a very steep dose response curve with an early plateau. In other words the key role of leptin is to signal starvation and loss of body energy reserves (no leptin) and discriminate this from adequate energy stores (some leptin). It appears not to follow that there are increasing effects of higher concentrations of leptin still, as in obesity. I suspect this is also true in the immune system. Perhaps as an emergency energy saving manoeuvre, energy burning by immune cells is turned down by complete leptin deficiency, and quickly goes back to normal when some leptin is given. So it might be best to view complete leptin

deficiency as a mildly immunocompromised state which is reset by replacing leptin. If this is true then the implication is that states in which leptin is extremely low may attenuate IBD activity. This will mostly be true in extremely thin, malnourished people. Is there any hint that disease severity reduces when extreme energy depletion and loss of all body fat occurs? I don't think there are any strong grounds for arguing that because some leptin vs no leptin may be bad for IBD, so increasing amounts of leptin are increasingly bad. The vague case that creeping fat in Crohn's may somehow fit in this picture (?locally high leptin concentrations influencing local disease activity) is highly speculative and unsupported by any data.

Given all this, I think that although the paper is interesting and attractive and definitely worthy of publishing, I am personally not convinced that it provides clear enough mechanistic evidence, enough novelty, or enough general interest to warrant publication in Nature Communications.

Point by point response

Reviewer #1:

“The Authors have satisfactorily addressed all the issues raised.”

We thank the referee for the positive evaluation of our manuscript.

Reviewer #4:

„The revisions in the manuscript and supplemental figures address the concerns raised in the review.”

We would like to thank the reviewer for the constructive comments and helpful suggestions.

Reviewer #5:

„The authors have properly addressed raised comments by Reviewer 2 and thereby substantially improved this work.“

We thank the reviewer for the kind appreciation of our manuscript.

Reviewer #6:

“I comment, as requested, on the likely general importance of this report, especially from a metabolic and lipodystrophy perspective, following on from previous reviewer 3, with whose views I essentially concur. This an interesting and detailed opportunistic study of the immunological consequences of recombinant human leptin therapy used in the context of acquired generalised lipodystrophy, previous liver transplantation, and Crohn’s disease, which I have not come across associated with LD before in either published or unpublished cases. In response to previous reviews referencing of prior literature has been improved, and the paper generally is well written and clear.”

We thank the reviewer for the positive comments on our revised paper.

“The cellular and immunological work up is quite impressive given a single case and limited material, but the major problem is, of course, that this is a single, very complicated case, leaving some doubt about generalisability of findings.”

We thank the referee for appreciating our immunological characterization of the AGLCD patient and we agree with the reviewer that it would be important to further validate our findings in additional patients with acquired generalized lipodystrophy and concomitant autoimmune disease, which is unfortunately almost impossible to perform due to the extreme rarity of these patients and the singularity of the AGLCD patient. However, we do believe that the existing evidence from mice underscores the pro-inflammatory potential of leptin in inflammatory bowel disease as previous studies have demonstrated that leptin-deficient *ob/ob* mice are protected from DSS-induced colitis (Siegmond et al., *Leptin: a pivotal mediator of intestinal inflammation in*

mice. Gastroenterology 2002) and that the pharmacologic blockade of the leptin receptor attenuates disease severity in dextran-sodium-sulfate (DSS) induced colitis in mice (Singh et al., *Leptin antagonist ameliorates chronic colitis in IL-10^{-/-} mice*. Immunobiology 2013). This pro-inflammatory role of leptin in intestinal inflammation is furthermore supported by the observation that leptin-receptor deficient CD4⁺ T cells fail to induce intestinal inflammation in transfer models of colitis (Reis et al., *Leptin receptor signaling in T cells is required for Th17 differentiation*. J Immunology 2015).

“The authors do a good job of placing their findings in the context of prior studies of immunological effects of leptin, but therein lies another problem. It has long been suggested that leptin deficiency is somewhat immunosuppressant, and that leptin reconstitution may reactivate quiescent autoimmune disease, but this has been backed only by a handful of clinical anecdotes, and by evaluation of effects of leptin in people without overt autoimmune disease. This means that the current case is of value, but equally that the information provided is also rather incremental.”

We thank the reviewer for pointing out that there have been single reports of possible aggravations of autoimmune disease under rLeptin substitution (Javor, E.D., et al. *Proteinuric nephropathy in acquired and congenital generalized lipodystrophy: baseline characteristics and course during recombinant leptin therapy*. J Clin Endocrinol Metab 2004). We concur with the reviewer's assessment that the available literature regarding the immune-stimulatory potential of rLeptin in humans remains vague and we therefore are convinced that our in-depth immune analysis might help to better understand how rLeptin substitution confers to auto-immunity. In our opinion, rleptin does not directly trigger inflammation (otherwise rLeptin should cause autoimmunity in all forms of lipodystrophy, which is not the case), but is likely enhancing autoimmunity by facilitating the production of pro-inflammatory cytokines such as TNFalpha and by regulating immune cell differentiation and cellular expansion of auto-reactive lymphocytes. We believe that our observations are not only important for the clinical management of potential side effects of rLeptin substitution in patients with generalized lipodystrophies. In our opinion, they also argue in favor of a broader role of leptin for proper immune cell function as leptin deficiency is associated with decreased and impaired NK cells both in lipodystrophy patients and leptin deficient *ob/ob* mice (Tian et al. *Impaired natural killer (NK) cell activity in leptin receptor deficient mice: leptin as a critical regulator in NK cell development and activation*. Biochem Biophys Res Commun, 2002).

We therefore agree with the reviewer that leptin deficiency should be considered as a cause for immune deficiency. Accordingly, patients with malnutrition and consecutive low levels of leptin suffer from an increased susceptibility for severe infections including leishmaniasis and amebiasis due to impaired T and NK cell functions (Maurya et al. *Leptin functions in infectious diseases*. Frontiers in Immunology 2018). *Vice versa*, Dayakar et al. recently described that leptin stimulation improves T cell responses in malnourished mice infected with *Leishmania donovani* ultimately resulting in decreased parasite loads in leptin treated mice (Dayakar et al. *Leptin regulates Granzyme-A, PD-1 and CTLA-4 expression in T cell to control visceral leishmaniasis in BALB/c mice*. Scientific Reports, 2017). Taken together, we think that our study underscores the pleiotropic functions that leptin possesses in the differentiation, function and metabolic programming of lymphocytes.

“Possible mechanisms underlying the observations are rationalised with reference to other studies, for example in db/db mice, but this study alone cannot discriminate direct and indirect effects of leptin. Leptin in LD reduces appetite, improves blood lipid, glucose and insulin levels, and in principle any of these may contribute to changes in immune cell function. Without controls such as those with severe insulin resistance who are leptin replete, those with poorly controlled insulin deficient diabetes, or those with absent leptin levels but no diabetes and insulin resistance (and these can be found) indirect effects of leptin cannot be ruled out.”

We thank the reviewer for this important comment, which we address in the revised discussion. We completely agree that the observed in-vivo effects of rLeptin substitution in the AGLCD patient likely consist of a summary of primary, secondary and tertiary effects of leptin including not only the leptin-mediated regulation of appetite, blood lipids, glucose and insulin levels but which might also include a leptin-dependent regulation of epithelial homeostasis in the gut and of the microbiota. However, previous work from our group as well as Reis and colleagues could already unambiguously show, that leptin has direct primary effects on the function and differentiation of lymphocytes in intestinal autoimmunity as CD4⁺ T cells from *Lepr^{fl/fl}-CD4-Cre* mice fail to induce or at least delay the onset of intestinal inflammation in transfer models of colitis due to a defective production of inflammation inducing cytokines (Siegmund et al. *Leptin receptor expression on T lymphocytes modulates chronic intestinal inflammation in mice*. Gut 2004; Reis et al., *Leptin receptor signaling in T cells is required for Th17 differentiation*. J Immunology 2015). We thank the reviewer for suggesting additional control groups to better delineate secondary confounding metabolic effects of leptin substitution. However, we think that the suggested control groups would be not suitable to distinguish between primary and secondary effects of rLeptin substitution since all of the mentioned patient-groups have fat tissue, which would make a comparison between the AGLCD and these groups obsolete since other adipokines such as ghrelin or adiponektin are also likely to affect the immune cell composition and differentiation in the suggested groups and would be absent in the AGLCD patient.

“Another question is the wider importance of a possible role of leptin in immune function in Crohn’s. In metabolic control leptin appears to have a very steep dose response curve with an early plateau. In other words the key role of leptin is to signal starvation and loss of body energy reserves (no leptin) and discriminate this from adequate energy stores (some leptin). It appears not to follow that there are increasing effects of higher concentrations of leptin still, as in obesity. I suspect this is also true in the immune system. Perhaps as an emergency energy saving manoeuvre, energy burning by immune cells is turned down by complete leptin deficiency, and quickly goes back to normal when some leptin is given. So it might be best to view complete leptin deficiency as a mildly immunocompromised state which is reset by replacing leptin. If this is true then the implication is that states in which leptin is extremely low may attenuate IBD activity. This will mostly be true in extremely thin, malnourished people. Is there any hint that disease severity reduces when extreme energy depletion and loss of all body fat occurs? I don’t think there are any strong grounds for arguing that because some leptin vs no leptin may be bad for IBD, so increasing amounts of leptin are increasingly bad. The vague case that creeping fat in Crohn’s may somehow fit in this picture (locally high leptin concentrations influencing local disease activity) is highly speculative and unsupported by any data.”

We thank the reviewer for this valid comment. As pointed out above, we agree that leptin deficient patient should be considered to be partially immunosuppressed, which is also in agreement with a higher susceptibility for infections in malnourished or cachexic patients, in

which leptin levels are low (Maurya et al. *Leptin functions in infectious diseases*. *Frontiers in Immunology* 2018). The referee's reasoning that malnourishment and subsequently low levels of leptin might therefore attenuate IBD is very interesting. In the literature we could only find one convincing publication comparing the serum levels of leptin in IBD patients with acute flare or patients in remission (UC). Thereby, UC patients had significantly higher levels of leptin in the acute phase of colitis when compared to patients in remission and the serum concentrations of leptin were directly correlated with expression levels of TNF α and IL-1 β in the serum (Biesiada et al. *Expression and release of leptin and proinflammatory cytokines in patients with ulcerative colitis and infectious diarrhea*, *Journal of Physiology and Pharmacology*, 2012).

However, we believe that these observations do not allow the assumption that malnourishment and consecutively low levels of leptin would positively affect inflammation in IBD patients because apart from reducing leptin, malnourishment potentially has plenty of additional effects that might interfere with disease activity in Crohn's disease: For example, malnourished patients were shown to have a leaky epithelial barrier, which subsequently triggers bacterial translocation ultimately leading to increased recruitment and activation of immune cells in the lamina propria (Welsh et al. *Gut barrier function in malnourished patients*. *Gut* 1998, Amadi et al. *Impaired barrier function and autoantibody generation in malnutrition enteropathy in Zambia*. *EbioMedicine* 2017). Remarkably, Paul et al. have previously described that leptin is elevated in creeping fat of Crohn's disease patients (Paul G, et al. *Profiling adipocytokine secretion from creeping fat in Crohn's disease*. *Inflamm Bowel Dis*, 2006). Accordingly, CD patients with a high burden of creeping fat show elevated levels of leptin in the serum and the amount of visceral fat accumulation/total fat mass (but not BMI!) is directly correlated to a higher disease activity (Büning et al. *Visceral adipose tissue in patients with Crohn's disease correlates with disease activity, inflammatory markers and outcome*. *Inflamm Bowel Dis* 2015). In line with this observation that creeping fat might serve as a local source for leptin and other proinflammatory cytokines and Coffey and colleagues have recently registered in a small retrospective study that a radical resection of the mesentery and creeping fat in CD patients increases the recurrence free survival of patients (Coffey et al. *Inclusion of the mesentery in ileocolic resection for Crohn's disease is associated with reduced surgical recurrence*, *Journal of Crohn's and Colitis* 2018). Our group has recently observed in mice, that intestinal inflammation and subsequent bacterial translocation into the mesenteric fat induces the local production of leptin by adipocytes in a TLR-dependent manner in DSS treated wildtype mice but not in My88^{-/-} mice (Batra et al. *Mesenteric fat-control site for bacterial translocation in colitis? Mucosal Immunology* 2012). Given the high infiltrations of creeping fat with lymphocytes such as CD8⁺ T cells and monocytes (Kredel et al. *T cell composition in ileal and colonic creeping fat-separating ileal from colonic Crohn's disease*. *Journal of Crohn's and colitis* 2019), we think it is plausible to argue that local leptin production will likely affect the differentiation as well as the functionality of fat residing lymphocytes especially in regard of the observation that T cells express leptin receptors and that *Lepr*-deficient T cells fail to induce colitis in mice (Siegmond et al. *Leptin receptor expression on T lymphocytes modulates chronic intestinal inflammation in mice*. *Gut* 2004; Reis et al., *Leptin receptor signaling in T cells is required for Th17 differentiation*. *J Immunology* 2015). However, we agree with the reviewer that additional studies are required to decipher the role of creeping fat in the pathogenesis of Crohn's disease and to discriminate the single roles of the various adipokines secreted by creeping fat. In light of this complex interplay between adipokines and immunity, we believe that the AGLCD patient represents a singular chance to look exclusively at the impact leptin and to blend out secondary effects of additional adipokines that are also elevated in the creeping fat of CD patients such as adiponectin.

“Given all this, I think that although the paper is interesting and attractive and definitely worthy of publishing, I am personally not convinced that it provides clear enough mechanistic evidence, enough novelty, or enough general interest to warrant publication in Nature Communications.”

We thank the reviewer for appreciating our study.